# Long-Term Maintenance of Sinus Rhythm Is Associated with Favorable Echocardiographic Remodeling and Improved Clinical Outcomes after Transcatheter Aortic Valve Replacement

**DOI:** 10.3390/jcm11051330

**Published:** 2022-02-28

**Authors:** Young Choi, Byung-Hee Hwang, Gyu-Chul Oh, Jin Jin Kim, Eunho Choo, Min-Chul Kim, Juhan Kim, Hae Ok Jung, Ho-Joong Youn, Wook-Sung Chung, Kiyuk Chang

**Affiliations:** 1Division of Cardiology, Department of Internal Medicine, Seoul St. Mary’s Hospital, College of Medicine, The Catholic University of Korea, Seoul 06591, Korea; superstar@catholic.ac.kr (Y.C.); hbhmac@naver.com (B.-H.H.); david.gyuchul@gmail.com (G.-C.O.); jjbluemoon@catholic.ac.kr (J.J.K.); cmcchu@catholic.ac.kr (E.C.); hojheart@catholic.ac.kr (H.O.J.); younhj@catholic.ac.kr (H.-J.Y.); chungws@catholic.ac.kr (W.-S.C.); 2Cardiovascular Research Institute for Intractable Disease, College of Medicine, The Catholic University of Korea, Seoul 06591, Korea; 3Department of Cardiology, Chonnam National University Hospital, Chonnam National University Medical School, Gwangju 61469, Korea; kmc3242@hanmail.net (M.-C.K.); kim@zuhan.com (J.K.)

**Keywords:** transcatheter aortic valve replacement, atrial fibrillation, aortic valve stenosis

## Abstract

Periprocedural atrial fibrillation (AF) is associated with poor prognosis after transcatheter aortic valve replacement (TAVR). We evaluated the impact of long-term sinus rhythm (SR) maintenance on post-TAVR outcomes. We enrolled 278 patients treated with TAVR including 87 patients with periprocedural AF. Patients with periprocedural AF were classified into the AF-sinus rhythm maintained (AF-SRM) group or the sustained AF group according to long-term cardiac rhythm status after discharge. Patients without AF before or after TAVR were classified into the SR group. The primary clinical outcome was a composite of all-cause death, stroke, or heart failure rehospitalization. The AF-SRM and the SR groups showed significant improvements in left ventricular ejection fraction and left atrial volume index at one year after TAVR, while the sustained AF group did not. During 24.5 (±16.1) months of follow-up, the sustained AF group had a higher risk of the adverse clinical event compared with the AF-SRM group (hazard ratio (HR) 4.449, 95% confidence interval (CI) 1.614–12.270), while the AF-SRM group had a similar risk of the adverse clinical event compared with the SR group (HR 0.737, 95% CI 0.285–1.903). In conclusion, SR maintenance after TAVR was associated with enhanced echocardiographic improvement and favorable clinical outcomes.

## 1. Introduction

Atrial fibrillation (AF) is a highly prevalent arrhythmia in elderly patients with structural heart disease [1]. The risk of AF rises in aortic valve stenosis (AS), which shares risk factors with AF and causes pressure overload in the left atrium (LA). Coexisting AF was observed in more than 1/3 of patients with severe AS in previous studies [2]. Transcatheter aortic valve replacement (TAVR) is becoming an increasingly popular treatment for patients with severe AS. The clinical outcome after TAVR has gradually improved, and its indication has been expanded; recent data demonstrated that TAVR can be a favorable treatment option even in patients with low surgical risks [3,4]. However, periprocedural AF has been shown to have a deleterious effect on hard outcomes in AS patients, even after successful treatment with TAVR [5,6]. Periprocedural AF is divided into two categories according to the timing of detection, pre-existing AF or new-onset AF (NOAF), and both are related to increased adverse clinical events after TAVR [7,8,9,10,11,12,13].

Maintenance of sinus rhythm (SR) can have beneficial effects in terms of mortality and stroke in AF patients [14,15]. Rhythm control of AF in patients with concomitant heart failure (HF) has been demonstrated to improve left ventricular function [16]. In addition, previous studies on post-operative AF after cardiac surgery suggest that the majority of NOAF is resolved after discharge [17]. Successful TAVR can reduce pressure overload of LA that may result in reduction of AF burden. However, there are limited data regarding the rhythm control success rate and impact of long-term SR maintenance on prognosis after TAVR in patients with periprocedural AF.

We investigated whether long-term SR maintenance in patients with periprocedural AF would affect echocardiographic improvement and clinical outcomes after TAVR using prospective registry data.

## 2. Materials and Methods

### 2.1. Study Population

All consecutive patients with severe AS who underwent TAVR between June 2015 and December 2018 at two cardiovascular centers were enrolled. Included patients had symptomatic severe AS, defined as an aortic valve area <1.0 cm^2^ with a mean aortic valve gradient of ≥40 mmHg or a peak aortic jet velocity of 4.0 m/s. Risk for surgery was evaluated using the Society of Thoracic Surgeons (STS) score [18] and the logistic European System for Cardiac Operative Risk evaluation (euroSCORE) method [19]. The treatment modality decision was made by a heart team including cardiac surgeons, interventional cardiologists, imaging specialists, and radiologists who considered age, comorbidities, performance status, comprehensive risk assessment by STS score, and anatomical suitability of cardiac, aortic, and major vascular structures. TAVR was performed using either the self-expandable Medtronic CoreValve (Medtronic, Inc., Minneapolis, MN, USA) or the balloon-expandable Edwards SAPIEN valve (Edwards Lifesciences, Irvine, CA, USA). Cardiac rhythm was continuously monitored using an in-hospital ECG telemonitoring system during the index hospitalization for TAVR. The study was approved by the Institutional Review Board of the Catholic Medical Center of Korea. All subjects provided written informed consent.

### 2.2. Definitions and Outcomes

Periprocedural AF was defined as any documented AF episode before TAVR or during index hospitalization after TAVR. AF was diagnosed when an episode of AF was detected by 12-lead ECG (electrocardiogram), or AF lasting for >30 s was detected on ECG telemonitor or Holter test. Patients who did not have a prior history of AF or documented AF during index hospitalization were classified to the SR group. Patients with periprocedural AF were divided into the two groups according to the long-term cardiac rhythm status after discharge; those with no evidence of recurrent AF until the last follow-up period from discharge were classified to the AF-sinus rhythm maintained (AF-SRM) group, and those with any documented AF episode after discharge were classified to the sustained AF group. The primary clinical endpoint was a composite of all-cause death, stroke, or HF events requiring rehospitalization. An HF event was confirmed when a patient had signs and symptoms of congestion requiring intravenous diuretics or emergent hemofiltration. The secondary clinical endpoint was each component of the primary endpoint and a major bleeding event as defined by the International Society on Thrombosis and Hemostasis [20]. The definition of pre-existing AF was documented AF before TAVR or patient-reported history of prior AF, and the definition of NOAF was newly diagnosed AF during index hospitalization after TAVR.

### 2.3. AF Management and Follow-Up

Patients with symptomatic AF were treated with antiarrhythmic drugs (AAD), and electrical cardioversion was considered when rhythm control of symptomatic AF was not successful after 1–2 days of medical treatment. Choice and the duration of AAD therapy were left to each physician’s discretion. Patients with AF and CHA_2_DS_2_-Vasc score of ≥2 were prescribed an anticoagulant unless an excessive bleeding risk was present. All patients were routinely followed up at 3, 6, and 12 months after TAVR and every 6 months thereafter. A 12-lead ECG was routinely checked at every follow-up visit, and a 24 h Holter test or event recording ECG was performed in patients with relevant symptoms. Transthoracic echocardiograms were performed before and 12 months after TAVR and every 12 months thereafter. Obtained echocardiographic data were analyzed in a Core Echo Laboratory in each medical center by experienced physicians. To validate the complete follow-up data, information related to censored data was additionally obtained from telephone interviews. All clinical outcomes of interest were confirmed and adjudicated by the clinical events committee of the Cardiovascular Center of Seoul St. Mary’s Hospital.

### 2.4. Statistical Analyses

Continuous variables are presented as mean ± standard deviation for normally distributed data or median (25th–75th percentiles) for non-normally distributed data. Normally distributed variables were compared using the Student’s *t*-test for two-tailed analysis and one-way analysis of variance (ANOVA) for three groups. Non-normally distributed variables were compared using the Mann–Whitney U-test for two-tailed analysis and the Kruskal–Wallis test for three groups. Categorical variables are presented as frequencies with percentages (%) and were compared by the chi-squared test or Fisher’s exact test. Cumulative incidences of primary endpoints were estimated by Kaplan–Meier survival curves and compared using log-rank tests. Multivariate Cox proportional hazard regression analysis with backward elimination including any variable with a *p*-value <0.10 in univariate analyses and relevant variables with known clinical significance was performed to adjust for differences in baseline variables between the groups. Cutoff value of left atrial volume index (LAVI) for the prediction of AF rhythm control was determined using receiver operating curve with the Youden index. Significance of echocardiographic changes at one year was tested using the Wilcoxon signed rank test. For two-tailed analyses, *p*-values <0.05 were considered statistically significant, and significance levels were adjusted to <0.017 using the Bonferroni correction for the comparison of three groups [21]. All statistical analyses were performed using R version 3.6.2 (R Foundation, Indianapolis, IN, USA).

## 3. Results

### 3.1. Baseline Characteristics

A total of 278 patients undergoing TAVR were enrolled. The mean patient age was 78.8 (± 6.7) years, and 128 (46.0%) patients were male. TAVR was successfully conducted without intraprocedural mortality in all subjects. A prosthetic valve was delivered by the trans-femoral approach in 274 (98.2%) patients, trans-aortic approach in 3 (1.1%), and trans-subclavian artery approach in 1 (0.4%). There were 87 patients with either pre-existing AF (*n* = 52) or NOAF (*n* = 35). Among the 87 patients with periprocedural AF, SR was maintained in 43 (49.4%) patients during the whole follow-up period after discharge. There were no significant differences in age, sex, body mass index, preprocedural aortic valve area, or the prescription rates of beta-blockers or renin–angiotensin system blockers among the SR, AF-SRM, and sustained AF groups (Table 1). Chronic kidney disease and prior MI were more prevalent in patients with AF. Patients with AF had higher CHA_2_DS_2_-Vasc scores, higher LAVI and lower left ventricular ejection fraction (LVEF) compared with the SR group. Most patients in the SR group were prescribed dual antiplatelet agents after TAVR (78.5%), while 70.1% of patients with AF were prescribed anticoagulants. Patients with sustained AF had significantly higher LA diameter (49.2 ± 6.9 vs. 45.3 ± 6.9 mm in the sustained AF and the AF-SRM group, respectively, *p* < 0.001) and higher LAVI (65.4 ± 18.9 vs. 52.1 ± 24.6 mL/m^2^, *p* < 0.001) compared with the AF-SRM group. AAD was used in 29 patients with periprocedural AF, and amiodarone was used in the majority of patients (24/29). Prescription rate for AAD was similar in the AF-SRM and the sustained AF groups.

### 3.2. Echocardiographic Improvement

Follow-up echocardiogram one year after TAVR was performed in 199 (71.5%) patients. Baseline LVEF was lower in the AF-SRM and sustained AF groups compared with the SR group, but follow-up LVEF at one year was similar between the SR and AF-SRM groups and was significantly lower in the sustained AF group (62.6 ± 5.9% vs. 62.0 ± 5.2% vs. 56.1 ± 8.3% in the SR, AF-SRM, and sustained AF groups, respectively; *p* < 0.001) (Table 2). LVEF was significantly increased over one year in the SR and AF-SRM groups, while change in LVEF was not significant in the sustained AF group (∆EF = 2.6 ± 9.2%, 6.1 ± 11.8%, and 2.0 ± 11.6%, in the SR, AF-SRM, and sustained AF groups, respectively; *p*-values compared to baseline: 0.001, 0.007, and 0.310) (Figure 1A–C). The proportion of patients with reduced LVEF (<50%) at baseline were 9.8%, 27.3%, and 26.5% in the SR, AF-SRM, and sustained AF groups, and decreased to 3.8%, 3.0%, and 20.5% at one year. Both the SR and the AF-SRM group showed significant reduction in LAVI at one year (∆LAVI = −6.3 ± 11.7 and −10.1 ± 16.0 mL/m^2^ in the SR and AF-SRM group; *p*-values compared to baseline: <0.001 and 0.002) (Figure 1D,E). However, baseline LAVI was markedly higher in the sustained AF group without a significant change at one year after TAVR (∆LAVI = −1.2 ± 16.6 mL/m^2^; *p*-value compared to baseline: 0.690) (Figure 1F). Right ventricular systolic pressure (RVSP) was significantly reduced after one year in all groups (∆RVSP = −4.2 ± 11.8, −9.6 ± 12.3, and −5.8 ± 14.2 mmHg in the SR, AF-SRM, and sustained AF groups, respectively; *p*-values compared to baseline: <0.001, <0.001, and 0.031).

### 3.3. Clinical Outcomes

The mean follow-up duration was 24.5 (± 16.1) months. The primary endpoint, a composite of all-cause death, stroke or rehospitalization due to HF, occurred in 30 (15.7%) patients in the SR group, 5 (11.6%) in the AF-SRM group, and 16 (36.3%) in the sustained AF group (Table 3). Incidence of the primary endpoint was significantly higher in patients with periprocedural AF (either AF-SRM or sustained AF) compared with the SR group (15.7% vs. 24.1% in patients with SR and AF, respectively; log-rank *p* = 0.038) (Figure 2A). However, risk of the primary endpoint in patients with AF-SRM was comparable to that in the SR group (AF-SRM group vs. SR group: hazard ratio (HR) 0.737, 95% confidence interval (CI) 0.285–1.903, *p* = 0.529) while the sustained AF group had significantly worse outcome (*p* = 0.001 compared to the SR group) (Figure 2B). Among the patients with AF, sustained AF after TAVR was associated with more than four times higher risk of the primary endpoint compared to the AF-SRM group (sustained AF vs. AF-SRM: HR 4.449, 95% CI 1.614–12.270, *p* = 0.004). Every component of the primary endpoint except stroke was numerically higher in the sustained AF group. HF rehospitalization did not occur in the AF-SRM group, while 10 (22.7%) patients in the sustained AF group experienced HF rehospitalization. A stroke event occurred in four (2.1%) patients in the SR group, one patient (2.3%) in the AF-SRM group, and no patient in the sustained AF group. The incidence of major bleeding was non-significantly higher in the sustained AF group compared with the AF-SRM group (HR 3.095, 95% CI 0.592–16.17, *p* = 0.181). In the SR group, new paroxysmal AF was detected after discharge in four patients at 3, 6, 12, and 15 months after TAVR. Despite appropriate medical therapy, one patient experienced stroke event and all four patients were expired with a median survival of 11.8 months after TAVR. In multivariate analysis, age, chronic kidney disease, and SR maintenance (either the SR or AF-SRM group) were identified as independent predictors for a favorable clinical outcome, while the presence of any AF without considering rhythm status were not (Table 4). Among patients with periprocedural AF, SR maintenance and anticoagulation were independent predictors for favorable clinical outcomes (Table 5).

### 3.4. Impact of SR Maintenance in Pre-Existing AF and NOAF

The median time to NOAF development from the procedure was 1 (0–2) day. SR was maintained in 19/52 (36.5%) patients with pre-existing AF and in 24/35 (68.5%) patients with NOAF. In patients with NOAF, long-term SRM was associated with a remarkable reduction of the primary endpoint (8.3% vs. 72.7% in the NOAF-SRM and sustained NOAF, respectively; *p* = 0.002) (Appendix A). Among each component of the primary endpoint, all-cause mortality and HF rehospitalization rate was significantly lower in SR maintained NOAF patients. In patients with pre-existing AF, incidence of the primary endpoint was non-significantly lower in the SRM group (15.8% vs. 24.2% in the SRM and sustained pre-existing AF group, *p* = 0.312). The success rate of pharmacologic rhythm control with or without electrical cardioversion was 56.2% for pre-existing AF and 61.5% for NOAF. Ten (19.2%) patients with pre-existing AF and 17 (48.5%) patients with NOAF were AF-free without any AAD after TAVR.

### 3.5. Predictors for SR Maintenance

SR maintenance rate was 40/54 (74.1%) in patients with paroxysmal AF and 3/33 (10.0%) in patients with persistent AF. In multivariate analysis, paroxysmal type AF and LAVI < 46 mL/m^2^ were significant predictors for SRM (Appendix A). These two variables had incremental value in the prediction of SR maintenance (SR maintenance rate: 4.3% for 0 predictor, 55.1% for either predictor, and 90.9% for two predictors) with a C-statistic of 0.867. Having more than one predictor variable (either paroxysmal AF or LAVI < 46 mL/m^2^) predicted SR maintenance with a sensitivity of 0.973 and a specificity of 0.594.

## 4. Discussion

We assessed the impact of SR maintenance in a prospective, dual-center registry of patients undergoing TAVR. We found that maintenance of SR was associated with enhanced echocardiographic improvement in terms of LVEF and LAVI during the 12 months following TAVR. SR maintenance was also associated with significant reduction in clinical events including death, stroke, or rehospitalization due to HF. While patients with AF were generally at higher risk of adverse clinical events after TAVR than patients without AF, periprocedural AF was not a predictor for worse prognosis when AF did not recur after discharge. In our data, SR maintenance was mostly observed in patients with paroxysmal AF and low LAVI.

In previous literature, both pre-existing AF and NOAF were shown to be associated with increased post-TAVR mortality, complications and rehospitalization due to HF, but the relationship with stroke and bleeding remains unclear [5,8,9,12,22,23]. In the PARTNER trial, the presence of AF before or after TAVR was associated with increased mortality, but AF was not associated with an increased risk of stroke or major bleeding [8]. A study using the FRANCE-2 registry including 3933 patients undergoing TAVR also showed that periprocedural AF was associated with higher mortality and morbidity without increasing the risk of stroke or major bleeding [9]. However, the SOURCE XT prospective multicenter registry, which included 2706 patients treated with the SAPIEN XT valve, showed that the risk of stroke was increased in patients with NOAF, as well as the risks of death, rehospitalization, and bleeding events [12]. In our study, a presence of either pre-existing AF or NOAF was related to postprocedural mortality and rehospitalization rates, but we first showed that the elevated risks were confined to the AF sustained group. We did not necessarily separate the AF group into pre-existing AF and NOAF because a distinctive diagnosis is often uncertain, especially in patients with paroxysmal AF. In addition, we observed significant differences in the changes in LVEF and LAVI after TAVR between the AF-SRM and sustained AF groups. Differences in echocardiographic remodeling can lead to clinical benefits, and it may have translated into the HF rehospitalization rate, which mostly drove the differences in the primary endpoint in our study.

Incidence of stroke after TAVR has been reported to be 3–10%, and periprocedural AF showed an adverse impact on the risk of stroke in previous literature [4,5,7,8,9,13]. In contrast, incidence of postprocedural stroke was relatively low in our study (1.8%), and periprocedural AF was not associated with higher risk of stroke. Risk of stroke after TAVR can be affected by various factors, including antithrombotic regimens. Patients with AF and high CHA_2_DS_2_-Vasc score who underwent TAVR are currently recommended to receive a single oral anticoagulant or combination of an anticoagulant with an antiplatelet agent [24]. However, prescription rates for anticoagulants were low or not precisely reported in previous studies [8,9,12,25]. An analysis from STS/ACC TVT registry data of 13,556 patients reported a 7.2% stroke rate among 1138 patients with NOAF; however, only 32.7% of patients with NOAF received oral anticoagulant at discharge despite a median CHA_2_DS_2_-Vasc score of 5 [25]. Recently, Seeger et al. showed long-term outcomes after TAVR in 345 SR patients and 272 AF patients. All patients with AF received either apixaban or warfarin, and the stroke rate at one year was only 1.5% without significant difference compared to patients with SR. Patients who were prescribed apixaban even showed a numerically lower incidence of stroke than patients prescribed warfarin (1.2% vs. 2.0%) [26]. A more recent study also showed low incidence of ischemic stroke (2.1%/year) in patients with AF undergoing TAVR who received appropriate oral anticoagulant therapy [27]. In a study of subjects with pacemakers, subclinical AF after TAVR with a lack of anticoagulation therapy has also been shown to be associated with an increased risk of stroke [28]. These data suggest that intensive AF screening and appropriate antithrombotic treatment can be beneficial in preventing post-TAVR stroke. In our study, patients were monitored using in-hospital continuous ECG telemetry after TAVR, and the detection rate of NOAF was higher (12.5%) than in previous studies (4.4–8.4%) [8,9,12,24]. According to CHA_2_DS_2_-Vasc scores, >70% (62/87) of patients with AF received oral anticoagulation, and 72.5% (45/62) of those receiving oral anticoagulation were given direct oral anticoagulants, which may have contributed to the low incidence of postprocedural stroke.

Although routine rhythm control strategy in patients with AF may not offer survival advantages, maintenance of SR was found to be an important determinant of better clinical outcome [15,29]. Recent prospective data (CASTLE-AF, CABANA trial) suggested that the benefit of rhythm control therapy on long-term survival is more pronounced in patients with AF and concomitant HF [16,30]. Although the CASTLE-AF study enrolled only patients with reduced LV systolic function, AF rhythm control would also benefit patients with severe AS with preserved LV ejection fraction, in which severe left ventricular diastolic dysfunction with systemic congestion is often present [31]. Moreover, TAVR would confer better circumstances for rhythm control in terms of LA pressure reduction. SR maintenance was a powerful predictor for improved clinical outcome in our study; although it is not clear whether the difference resulted from the patient characteristics or rhythm status itself, enhanced long-term echocardiographic improvement after TAVR in the SR maintained group would be a possible mechanism for the observed benefit. In the current analysis, SR maintenance was shown in 36.5% of pre-existing AF and 68.5% of NOAF, and long-term AAD was not required to maintain SR in 30.6% of all AF patients. In line with our study, a recent study showed that early postoperative AF following surgical aortic valve replacement or TAVR was not associated with worse outcomes, while post-discharge AF was a significant indicator of worse prognosis [32]. Additionally, our study showed that AF type and baseline LAVI can be used in conjunction to select a subgroup of AF patients in whom rhythm control strategy would be preferred after TAVR. However, although adjunctive electrical cardioversion raises the possibility of SR maintenance in patients with persistent AF, electrical cardioversion was not frequently performed in our study. Rhythm control success rate for persistent AF should be further evaluated in a future study with a well-specified rhythm control protocol including various treatment options.

### Limitations

In our study, the number of AF patients was small, which negatively affects statistical power. Although our study showed the benefit of SR maintenance on the primary clinical outcome, the impact of SR maintenance on mortality and stroke was not clearly demonstrated. This was a retrospective analysis, and the patient group was not classified by the received treatment; although we conducted multivariate regression analysis, the effect of differences in baseline covariates on study outcomes cannot be fully adjusted. Asymptomatic recurrence of paroxysmal AF after discharge could have been underdiagnosed in our study because routine continuous rhythm monitoring was not performed in patients without relevant symptoms of AF. Thus, it is possible that the SR maintenance rate was overestimated. Because the treatment strategy of AF was not unified, we cannot recommend any therapeutic option for AF based on our data. Finally, our study enrolled only Asian patients and the results may not be generalized to other ethnic groups.

## 5. Conclusions

We analyzed the impact of SR maintenance in patients with periprocedural AF on long-term outcomes after TAVR in a dual-center, prospective registry. Patients with AF who achieved long-term maintenance of SR showed favorable echocardiographic remodeling and improved clinical outcomes that were comparable to patients without AF, while sustained AF after TAVR was associated with higher risk of death, stroke, or rehospitalization and blunted echocardiographic improvement. Periprocedural AF is currently regarded as a worsening factor for prognosis after TAVR, but our study suggests that the impact of long-term rhythm status should also be considered. Further study is warranted to investigate optimal rhythm control strategy in patients with AF undergoing TAVR.

## Figures and Tables

**Figure 1 jcm-11-01330-f001:**
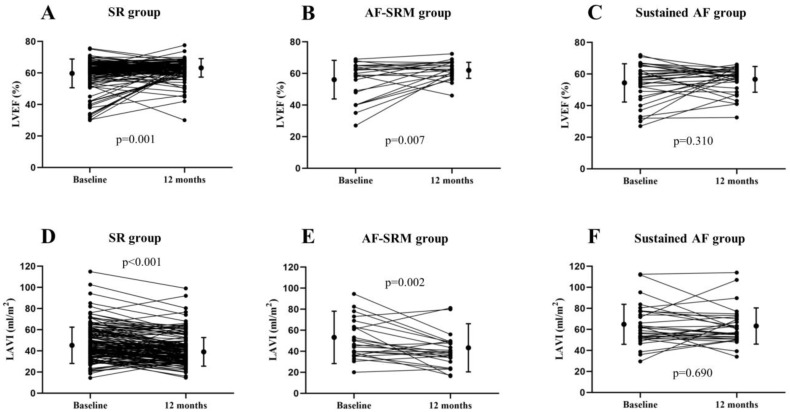
Changes in echocardiographic parameters over one year. LVEF changes in each individual in the SR group (**A**), the AF-SRM group (**B**), and the sustained AF group (**C**). LAVI changes in each individual in the SR group (**D**), the AF-SRM group (**E**), and the sustained AF group (**F**). SR = sinus rhythm; AF = atrial fibrillation; SRM = sinus rhythm maintained; LVEF = left ventricular ejection fraction; LAVI = left atrial volume index.

**Figure 2 jcm-11-01330-f002:**
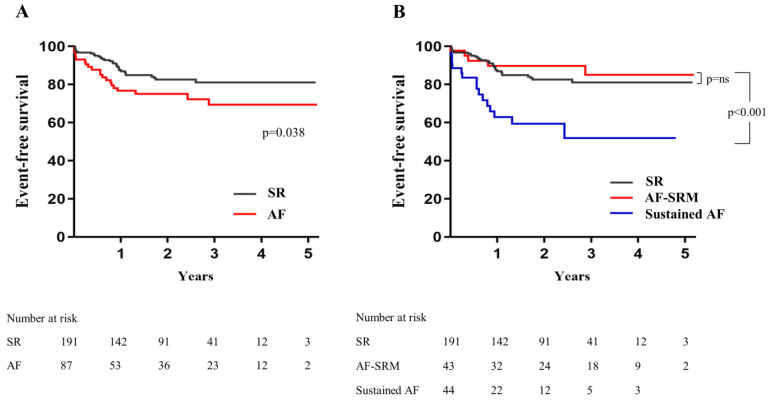
Freedom from the composite of all-cause death, stroke or rehospitalization due to heart failure. Comparison between the SR and the AF groups (**A**), comparison between the SR, AF-SRM, and sustained AF groups (**B**). SR = sinus rhythm; AF = atrial fibrillation; SRM = sinus rhythm maintained.

**Table 1 jcm-11-01330-t001:** Baseline characteristics and echocardiographic data in the three groups.

	SR (*n* = 191)	AF-SRM (*n* = 43)	Sustained AF (*n* = 44)	*p*
Age, years	78.4 ± 6.9	80.1 ± 5.9	79.3 ± 6.5	0.249
Male sex	89 (46.5%)	21 (48.8%)	18 (40.9%)	0.731
BMI, kg/m^2^	24.4 ± 3.9	24.3 ± 4.1	24.2 ± 3.9	0.707
Prosthetic valve type				0.024
Self-expandable	76 (39.8%)	24 (55.8%)	12 (27.3%)	
Balloon-expandable	115 (60.2%)	19 (44.2%)	32 (72.7%)	
Comorbidities, *n* (%)				
Hypertension	133 (69.6%)	32 (74.4%)	34 (77.3%)	0.541
Diabetes	58 (30.3%)	15 (34.9%)	13 (29.5%)	0.826
COPD	56 (29.3%)	12 (27.9%)	17 (36.4%)	0.616
Prior stroke	24 (12.5%)	8 (18.6%)	9 (20.5%)	0.305
PAD	32 (16.7%)	6 (14.0%)	9 (20.5%)	0.717
Prior MI	11 (5.8%)	6 (14.0%)	7 (15.9%)	0.039
Obstructive CAD	74 (38.7%)	19 (44.2%)	21 (47.7%)	0.495
CKD	21 (10.9%)	8 (18.6%)	13 (29.5%)	0.006
Creatinine, mg/dL	1.3 ± 1.7	1.3 ± 0.9	1.5 ± 1.3	0.638
CHA_2_DS_2_-Vasc score	4.0 ± 1.3	4.4 ± 1.6	4.5 ± 1.6	0.018
AVA, cm^2^	0.73 ± 0.17	0.69 ± 0.19	0.70 ± 0.19	0.356
AV Vmax, m/s	4.6 ± 0.6	4.8 ± 0.7	4.4 ± 0.7	0.093
LA diameter, mm	43.4 ± 5.4	45.3 ± 6.9	49.2 ± 6.9	<0.001
LAVI, mL/m^2^	46.3 ± 17.4	52.1 ± 24.6	65.4 ± 18.9	<0.001
LVEF, %	60.9 ± 9.9	57.1 ± 13.0	55.4 ± 11.9	0.001
STS score	8.1 ± 6.5	9.7 ± 8.9	9.3 ± 6.4	0.171
DC cardioversion, *n* (%)		3 (7.0%)	4 (9.1%)	0.901
Medications after TAVR, *n* (%)				
Single antiplatelet	12 (6.3%)	1 (2.3%)	0	0.150
Dual antiplatelet	150 (78.5%)	18 (41.9%)	5 (11.4%)	<0.001
Single OAC	23 (12.0%)	19 (44.2%)	25 (56.8%)	<0.001
Antiplatelet + OAC	2 (1.0%)	5 (11.6%)	12 (27.3%)	<0.001
Beta-blocker	53 (27.7%)	7 (16.3%)	10 (22.7%)	0.270
RAS blocker	84 (44.0%)	15 (34.9%)	17 (38.6%)	0.529
Antiarrhythmic drug		16 (37.2%)	13 (29.5%)	0.364

Categorical variables are presented as number (percentages) and continuous variables are presented as mean ± standard deviation. *p* < 0.017 indicates statistical significance. SR = sinus rhythm; AF = atrial fibrillation; SRM = sinus rhythm maintained; BMI = body mass index; COPD = chronic obstructive lung disease; PAD = peripheral artery disease; MI = myocardial infarction; CAD = coronary artery disease; CKD = chronic kidney disease; AVA = aortic valve area; AV = aortic valve; LA = left atrium; LAVI = left atrial volume index; LVEF = left ventricular ejection fraction; TAVR = transcatheter aortic valve replacement; OAC = oral anticoagulant; RAS = renin–angiotensin system.

**Table 2 jcm-11-01330-t002:** Echocardiographic parameters at one year after TAVI in the three groups.

	SR (*n* = 132)	AF-SRM (*n* = 33)	Sustained AF (*n* = 34)	*p*
LVEF, %	62.6 ± 5.9	62.0 ± 5.2	56.1 ± 8.3	<0.001
∆LVEF, %	2.6 ± 9.2	6.1 ± 11.8	2.0 ± 11.6	0.381
LAVI, mL/m^2^	39.4 ± 13.6	43.4 ± 23.2	62.6 ± 17.1	<0.001
∆LAVI, mL/m^2^	−6.3 ± 11.7	−10.1 ± 16.0	−1.2 ± 16.6	0.233
RVSP, mmHg	30.7 ± 8.8	28.7 ± 7.3	38.0 ± 15.3	0.004
∆RVSP, mmHg	−4.2 ± 11.8	−9.6 ± 12.3	−5.8 ± 14.2	0.221
AV Vmax, m/s	2.4 ± 0.4	2.5 ± 0.7	2.3 ± 0.5	0.511
AVA, cm^2^	1.6 ± 0.4	1.7 ± 0.5	1.6 ± 0.5	0.694
≥Moderate PVL, *n* (%)	11 (8.3%)	3 (9.4%)	0	0.205

*p* < 0.017 indicates statistical significance. ∆LVEF, ∆LAVI, and ∆RVSP are the changes at one year compared to preprocedural data. SR = sinus rhythm; AF = atrial fibrillation; SRM = sinus rhythm maintained; LVEF = left ventricular ejection fraction; LAVI = left atrial volume index; RVSP = right ventricular systolic pressure; AV = aortic valve; AVA = aortic valve area; PVL = perivalvular leakage.

**Table 3 jcm-11-01330-t003:** The primary and the secondary endpoints in the three groups.

	SR	AF-SRM	Sustained AF	HR *	95% CI	*p*
Death, stroke or rehospitalization	30 (15.7%)	5 (11.6%)	16 (36.4%)	4.449	1.614–12.270	0.004
All-cause death	20 (10.5%)	4 (9.3%)	7 (15.9%)	1.918	0.561–6.561	0.299
Cardiovascular death	5 (2.6%)	2 (4.7%)	5 (11.4%)	2.615	0.506–13.490	0.251
HF rehospitalization	11 (5.8%)	0	10 (22.7%)			<0.001
Stroke	4 (2.1%)	1 (2.3%)	0			0.616
Major bleeding	12 (6.3%)	2 (4.7%)	5 (11.4%)	3.095	0.592–16.17	0.181

*p* < 0.05 indicates statistical significance. * HR for the sustained AF group compared to the AF-SRM group. SR = sinus rhythm; AF = atrial fibrillation; SRM = sinus rhythm maintained; HR = hazard ratio; CI = confidence interval; HF = heart failure.

**Table 4 jcm-11-01330-t004:** Univariate and multivariate analysis for the prediction of the primary clinical endpoint (composite of all-cause death, stroke or rehospitalization due to heart failure) in the entire study population.

Variable	Crude HR	95% CI	*p*	Adjusted HR	95% CI	*p*
Age	1.073	1.031–1.122	0.002	1.098	1.041–1.158	<0.001
Male sex	1.205	0.693–2.071	0.525	1.124	0.545–2.317	0.750
LVEF < 50%	1.613	0.814–3.221	0.175	1.701	0.704–4.108	0.237
LAVI	1.021	1.012–1.034	0.002	1.011	0.996–1.027	0.142
Hypertension	2.162	1.014–4.596	0.046	1.811	0.717–4.575	0.208
Diabetes	1.834	0.922–3.661	0.086			
Stroke	0.851	0.386–1.885	0.685			
Obstructive CAD	1.819	1.054–3.171	0.033	1.589	0.805–3.135	0.181
CKD	3.642	2.016–6.595	<0.001	3.873	1.759–8.530	<0.001
Prior MI	1.568	0.671–3.684	0.300	0.749	0.239–2.342	0.619
CHA_2_DS_2_-Vasc score	1.154	0.963–1.382	0.141			
Use of balloon-expandable device	1.033	0.591–1.817	0.917	0.962	0.471–1.965	0.916
Anticoagulation	1.021	0.565–1.838	0.960	0.381	0.107–1.347	0.134
Any AF	1.809	1.032–3.163	0.038	0.343	0.088–1.341	0.124
SR maintenance	0.292	0.162–0.523	<0.001	0.081	0.018–0.368	0.001

HR and *p*-value were calculated using Cox regression analysis. *p* < 0.05 indicates statistical significance. AF = atrial fibrillation; HR = hazard ratio; CI = confidence interval; LVEF = left ventricular ejection fraction; LAVI = left atrial volume index; CAD = coronary artery disease; CKD = chronic kidney disease; MI = myocardial infarction; SR = sinus rhythm.

**Table 5 jcm-11-01330-t005:** Univariate and multivariate analysis for the prediction of the primary clinical endpoint in patients with periprocedural AF.

Variable	Crude HR	95% CI	*p*	Adjusted HR	95% CI	*p*
Age	1.044	0.975–1.117	0.312	1.078	0.988–1.176	0.090
Male sex	1.442	0.611–3.393	0.407	1.913	0.593–6.163	0.277
LVEF < 50%	1.303	0.504–3.376	0.593	1.275	0.302–5.378	0.739
LAVI	1.013	0.991–1.032	0.360	1.000	0.971–1.029	0.981
Hypertension	3.166	0.735–13.606	0.123			
Diabetes	1.202	0.472–3.114	0.702			
Stroke	0.740	0.221–2.513	0.625			
Obstructive CAD	1.425	0.623–3.497	0.383			
CKD	1.951	0.780–4.864	0.151	1.543	0.410–5.802	0.520
Prior MI	1.764	0.642–4.806	0.271	1.102	0.238–5.092	0.900
CHA_2_DS_2_-Vasc score	1.101	0.851–1.423	0.474			
Use of balloon-expandable device	0.828	0.359–1.941	0.654	1.128	0.324–3.920	0.849
Anticoagulation	0.685	0.284–1.650	0.395	0.104	0.015–0.720	0.021
Persistent AF	1.392	0.571–3.413	0.466	0.390	0.086–1.760	0.221
Pre-existing AF*	0.893	0.375–2.118	0.792	1.909	0.434–8.389	0.391
SR maintenance	0.220	0.081–0.626	0.004	0.023	0.003–0.183	<0.001

HR and *p*-value were calculated using Cox regression analysis. *p* < 0.05 indicates statistical significance. AF = atrial fibrillation; HR = hazard ratio; CI = confidence interval; LVEF = left ventricular ejection fraction; LAVI = left atrial volume index; CAD = coronary artery disease; CKD = chronic kidney disease; MI = myocardial infarction; SR = sinus rhythm.

## Data Availability

The datasets analyzed in this study are not publicly available but are available from the corresponding author on reasonable request.

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
