# Peer review of "Long-Term Maintenance of Sinus Rhythm Is Associated with Favorable Echocardiographic Remodeling and Improved Clinical Outcomes after Transcatheter Aortic Valve Replacement"

_jcm, 2022, doi:10.3390/jcm11051330_

Round 1
Reviewer 1 Report
This is an original research article, which aims to investigate the possible favorable effects of long-term sinus rhythm maintenance on echocardiographic remodeling parameters, such as left ventricular eject fraction and left atrial volume index, and on clinical outcomes, such as all-cause death, stroke, or heart failure rehospitalization, in patients with periprocedural AF after transcatheter aortic valve replacement.
Generally, the authors have in depth knowledge of this topic, and they have used the appropriate methodology, study design, and sufficient statistical analysis. The results are well-presented, clear and easy to understand, so as to reach safe and solid conclusions. Overall, the manuscript is well written and structured. Thus, I think it would make a nice addition to Journal of Clinical Medicine as an original research article.
However, the following points should be considered. I also suggest the following minor modifications at specific parts of this manuscript.
1)This study has some limitations with regard to the small number of patients undergoing TAVR, who were enrolled in this prospective, dual-center registry. Small study population negatively affects statistical power. As a result, the observed benefit of SR maintenance on mortality and stroke was deprived of statistical significance. On the other hand, a 24-h Holter test was performed only in symptomatic patients. In case it would be applied in all the participants, more precise conclusions could be reached. Moreover, only 71.5% of the patients were followed up with echocardiogram one year after TAVR, implying a significant number of patients lost during the follow-up period.
2)Number of references is quite small. Please include much more, as well recent literature, presenting more updated data and, thus, enhancing the quality of the manuscript.
3)Consider rewriting the following parts making the content clearer and easily comprehensible for the reader. Line 77-80: clarify more the subgroups.
4)Please refer the number of patients with contaminant HF before and after follow-up period in each sub-group category, if available.
5)Most patients with sustained AF had a balloon-expandable valve. Is there any correlation between the valve type and the outcomes in each subgroup category?
6) Line 147,171: P<0.017 indicates statistical significance. How did you end up in this statement? Please describe more the statistical procedure in the respective part of methods.
Author Response
We are very much grateful for sending us reviewer’s comments on our manuscript. Also we are very much thankful to reviewers who raised the issues on our manuscript which will help us upgrade our manuscript. we made every effort to modify our manuscript according to the reviewer’s opinions and you will find justified answers with revised text following each question. We thank you in advance for your consideration.
1)This study has some limitations with regard to the small number of patients undergoing TAVR, who were enrolled in this prospective, dual-center registry. Small study population negatively affects statistical power. As a result, the observed benefit of SR maintenance on mortality and stroke was deprived of statistical significance. On the other hand, a 24-h Holter test was performed only in symptomatic patients. In case it would be applied in all the participants, more precise conclusions could be reached. Moreover, only 71.5% of the patients were followed up with echocardiogram one year after TAVR, implying a significant number of patients lost during the follow-up period.
->Thank you for the comments. We agree that there are significant limitations in the study number and inconsistent Holter test results. Unfortunately, 17% of patients were loss to follow-up at one year – it was hard to avoid because our center is a tertiary hospital in which the majority of patients were referred from distant area. We would appreciate if you could understand that these limitations are difficult for us to solve at this point. We more precisely describe the limitations in the discussion.
After revision: Page 11-Limitation
In our study, the number of AF patients was small which negatively affects statistical power. Although our study showed the benefit of SR maintenance on the primary clinical outcome, the impact of SR maintenance on mortality and stroke was not clearly demonstrated. This was a retrospective analysis and the patient group was not classified by the received treatment; although we conducted multivariate regression analysis, the effect of differences in baseline covariates on study outcomes cannot be fully adjusted. Asymptomatic recurrence of paroxysmal AF after discharge could have been underdiagnosed in our study because routine continuous rhythm monitoring was not performed in patients without relevant symptom of AF. Thus, it is possible that the SR-maintenance rate was overestimated. Because the treatment strategy of AF was not unified, we cannot recommend any therapeutic option for AF based on our data. Finally, our study enrolled only Asian patients and the results may not be generalized to other ethnic groups.
2)Number of references is quite small. Please include much more, as well recent literature, presenting more updated data and, thus, enhancing the quality of the manuscript.
-> We added five references according to your comment, including 4 recent literatures regarding periprocedural AF in patients undergoing TAVR. A recent study extended from PARTNER-3 trial reported that not in-hospital postoperative AF, but post-discharge AF was associated with worse outcomes after TAVR, which is in line with our study results (ref. 1). Other references also showed recent data regarding the impact of AF on post-TAVR outcomes.
Added references:
- Shahim, B.; Malaisrie, S.C.; George, I.; Thourani, V.H.; Biviano, A.B.; Russo, M.; Brown, D.L.; Babaliaros, V;Guyton, R.A.; Kodali, S.K., et al. Postoperative atrial fibrillation or flutter following transcatheter or surgical aortic valve replacement. J Am Coll Cardiol Intv 2021, 14, 1565-1574
- Geisler, D.; Rudzinski, P.N.; Hasan, W.; Andreas, M.; Hasimbegovic, E.; Adlbrecht, C.; Winkler, B.; Weiss, G.; Strouhal, A.; Dell-Karth, G., et al. Identifying patients without a survival benefit following transfemoral and transapical transcatheter aortic valve replacement. J Clin Med 2021, 10, 4911.
- Van Mieghem, N.M.; Unverdorben, M.; Hengstenberg, C.; Mollmann, H.; Mehran, R.; Lopez-Otero, D.; Nombela-Franco, L.; Moreno, R.; Nordbeck, P.; Thiele, H., et al. Edoxaban versus vitamin K antagonist for atrial fibrillation after TAVR. N Engl J Med 2021, 385, 2150-2160.
- Khan, M.Z.; Zahid, S.; Khan, M.U.; Kichloo, A.; Jamal, S.; Minhas A.M.K.; Ullah, W.; Sattar, Y.; Mir, T.; Balla, S., et al. Outcomes of transcatheter aortic valve replacement in patiens with and without atrial fibrillation: insights from national inpatient sample. Expert Rev Cardiovasc Ther 2021, 19, 939-946.
Added descriptions in discussion
(Page 10) More recent study also showed low incidence of ischemic stroke (2.1%/year) in patients with AF undergoing TAVR who received NOAC therapy (ref. 3)
(Page 10) In previous literatures, both pre-existing AF and NOAF were shown to be associated with increased post-TAVR mortality, complications and rehospitalization due to HF, but the relationship with stroke and bleeding remains unclear (ref. 2,4).
(Page 11) In line with our study, recently study showed that early postoperative AF following surgical aortic valve replacement or TAVR was not associated with worse outcomes, while post-discharge AF was a significant indicator of worse prognosis (ref. 1)
3)Consider rewriting the following parts making the content clearer and easily comprehensible for the reader. Line 77-80: clarify more the subgroups.
->We corrected the method section to be more precise and comprehensible.
Before, Page 2, Materials and Methods – definitions and outcomes
AF was diagnosed when an episode of AF was detected by 12-lead ECG (electrocardiogram), or AF lasting for >30 seconds was detected on ECG telemonitor or Holter test. Pre-existing AF was defined as any documented AF episode before TAVR or patient-reported history of prior AF. NOAF was defined as newly diagnosed AF during index hospitalization after TAVR. Among patients with either pre-existing or NOAF, those with no evidence of AF until the last follow up period from discharge were classified to the AF-sinus rhythm maintained (AF-SRM) group, and those with documented AF at any time after discharge were classified to the sustained AF group.
After revision, Page 2, Materials and Methods – definitions and outcomes
Periprocedural AF was defined as any documented AF episode before TAVR or during index hospitalization after TAVR. AF was diagnosed when an episode of AF was detected by 12-lead ECG (electrocardiogram), or AF lasting for >30 seconds was detected on ECG telemonitor or Holter test. Patients who did not have a prior history of AF or documented AF during index hospitalization were classified to the SR group. Patients with periprocedural AF were divided into the two groups according to the long-term cardiac rhythm status after discharge; those with no evidence of recurrent AF until the last follow-up period from discharge were classified to the AF-sinus rhythm maintained (AF-SRM) group, and those with any documented AF episode after discharge were classified to the sustained AF group.
4)Please refer the number of patients with contaminant HF before and after follow-up period in each sub-group category, if available.
->Thank you for the comment. We added the proportion of HF-reduced EF patients in each group in the result section.
After revision –added description (Page 5, Result – echocardiographic improvement)
The proportion of patients with reduced LVEF (<50%) at baseline were 9.8%, 27.3%, and 26.5% in the SR, AF-SRM, and sustained AF groups, and decreased to 3.8%, 3.0%, and 20.5% at one year.
5)Most patients with sustained AF had a balloon-expandable valve. Is there any correlation between the valve type and the outcomes in each subgroup category?
->Like your comment, the type of prosthetic valve was different in the AF-SRM and the sustained AF groups. So we evaluated the association between the prosthetic valve type and clinical outcomes in the entire population (Table 4) and in the AF population (Table 5). The use of balloon-expandable valve was not significantly associated with the primary clinical endpoint in both the entire population (HR 1.033, p=0.917) and AF population (HR 0.828, p=0.654). Also the type of prosthetic valve was not an independent predictor for clinical outcome in multivariate analyses.
6) Line 147,171: P<0.017 indicates statistical significance. How did you end up in this statement? Please describe more the statistical procedure in the respective part of methods.
->In Table 1 and Table 2, three groups were compared, so we applied more strict statistical cutoff value using Bonferroni correction for multiple comparison analyses (ref). We described this statistical procedure in the method-statistical analyses section.
Ref) Dunnett, C.W.; A multiple comparisons procedure for comparing several treatments with a control. J Am Stat Assoc 1955, 50, 1096-1121.
Page 3, method – statistical analyses
For two-tailed analyses, p values <0.05 were considered statistically significant, and significance levels were adjusted to <0.017 using the Bonferroni correction for the comparison of three groups [21]
Reviewer 2 Report
Choi and colleagues present an interesting analysis concerning the prognostic impact of long term rhythm status among patients undergoing TAVI.
The main finding is that sinus rhythm maintenance is associated with more favorable clinical outcomes and structural remodeling, as compared with sustained atrial fibrillation. The results may motivate a more aggressive rhythm control approach among patients undergoing TAVI and with atrial fibrillation.
However, I have several major concerns that may potentially offset the value of this work:
1) Holter monitors or other long-term rhythm monitors were only used in case of symptoms, with clear implications for the detection of subclinical atrial fibrillation recurrences;
2) throughout the paper, the classification of atrial fibrillation proposed by the European Society of Cardiology (first-diagnosed, paroxysmal, persistent, and permanent) is not used. It would be interesting to have outcomes reported according to the subtype of atrial fibrillation, given the very different clinical meaning of the various types of the arrhythmia;
3) in the methods section, it is stated that continuous variables are reported as mean and standard deviation. Continuous variables should be checked for normality, and reported as mean (standard deviation) if normally distributed, or as median (1st-3rd quartile) if not normally distributed.
4) the paper enrolled Asian patients, and results may not be generalized to European/American settings.
Minor: page 2: "prospective registry data" should replace "a prospective registry data"
page 2: "underwent" should replace"were undergoing"
page 6: "HF rehospitalization did not occur" should replace "HF rehospitalization was not occurred"
Author Response
We are very much grateful for sending us reviewer’s comments on our manuscript. Also we are very much thankful to reviewers who raised the issues on our manuscript which will help us upgrade our manuscript. we made every effort to modify our manuscript according to the reviewer’s opinions and you will find justified answers with revised text following each question. We thank you in advance for your consideration.
Reviewer comment 2
1) Holter monitors or other long-term rhythm monitors were only used in case of symptoms, with clear implications for the detection of subclinical atrial fibrillation recurrences;
->Thank for the comments. We totally agree that absence of Holter results in asymptomatic patients is a main limitation of our study. It means that asymptomatic patients with paroxysmal AF recurrence may be included in the AF-SRM group. We described that limitation in detail in the discussion-limitation section.
Discussion-limitation (page 11)
Asymptomatic recurrence of paroxysmal AF after discharge could have been underdiagnosed in our study because routine continuous rhythm monitoring was not performed in patients without relevant symptom of AF. Thus, it is possible that the SR-maintenance rate was overestimated.
2) throughout the paper, the classification of atrial fibrillation proposed by the European Society of Cardiology (first-diagnosed, paroxysmal, persistent, and permanent) is not used. It would be interesting to have outcomes reported according to the subtype of atrial fibrillation, given the very different clinical meaning of the various types of the arrhythmia;
->In our study, the first-diagnosed AFs are mostly new-onset AF (NOAF) after TAVR, and the impact of preexisting vs. NOAF on outcome was analyzed separately (Table 5). So we thought it would not be relevant to use the “first-diagnosed AF” concept in our study. The definition of permanent AF is a sustained AF which is accepted by the patient and physician, and no further attempts for rhythm control will be undertaken. In our study, the patients were mostly referred for aortic valve disease, and they did not have thorough consultation regarding AF rhythm-control therapy with arrhythmia specialists. Also if the patients had long-standing persistent AF, AF rhythm control may be easier after resolution of aortic stenosis. Therefore, every patient with AF in our study could be considered as potential candidates for AF rhythm control after TAVR. So we thought the term “permanent AF” may not be suitable in our study, and all AF type was classified to either paroxysmal AF or persistent AF according to AF sustenance (7 days).
In the result section (3.4. predictors for SR maintenance), we described sinus rhythm maintenance rate according to AF type (74.1% in paroxysmal AF and 10.0% in persistent AF). The association between the clinical outcomes and AF type (Persistent vs. paroxysmal) are shown in Table 5 (persistent AF: HR1.392 [0.571 – 3.413], p=0.466). Unlike the sinus rhythm maintenance variable, AF type was not associated with clinical outcomes in univariate and multivariate analyses.
3) in the methods section, it is stated that continuous variables are reported as mean and standard deviation. Continuous variables should be checked for normality, and reported as mean (standard deviation) if normally distributed, or as median (1st-3rd quartile) if not normally distributed.
->Thank you for the comment. We checked the normality for continuous variables, and revised the method-statistical analysis section according to your comment.
After revision (page 3, statistical analyses)
Continuous variables are presented as mean ± standard deviation for normally distributed data or median (25th – 75th percentiles) for non-normally distributed data.
4) the paper enrolled Asian patients, and results may not be generalized to European/American settings.
-->We added the description in the discussion – limitation section.
After revision: limitation (page 11)
Finally, our study enrolled only Asian patients and the results may not be generalized to other ethnic groups.
Minor: page 2: "prospective registry data" should replace "a prospective registry data"
->We corrected the sentence according to your comment.
page 2: "underwent" should replace"were undergoing"
->We corrected the sentence according to your comment
page 6: "HF rehospitalization did not occur" should replace "HF rehospitalization was not occurred"
->We corrected the sentence according to your comment